# Maneuvering Target Tracking Using Simultaneous Optimization and Feedback Learning Algorithm Based on Elman Neural Network

**DOI:** 10.3390/s19071596

**Published:** 2019-04-02

**Authors:** Huajun Liu, Liwei Xia, Cailing Wang

**Affiliations:** 1School of Computer Science and Engineering, Nanjing University of Science and Technology, Nanjing 210014, China; louis_nust@163.com; 2Robotics Institute, Carnegie Mellon University, Pittsburgh, PA 15217, USA; wangcl@njupt.edu.cn; 3School of Automation, Nanjing University of Posts and Telecommunications, Nanjing 210023, China

**Keywords:** Elman neural network, maneuvering target tracking, simultaneous optimization and feedback learning

## Abstract

Tracking maneuvering targets is a challenging problem for sensors because of the unpredictability of the target’s motion. Unlike classical statistical modeling of target maneuvers, a simultaneous optimization and feedback learning algorithm for maneuvering target tracking based on the Elman neural network (ENN) is proposed in this paper. In the feedback strategy, a scale factor is learnt to adaptively tune the dynamic model’s error covariance matrix, and in the optimization strategy, a corrected component of the state vector is learnt to refine the final state estimation. These two strategies are integrated in an ENN-based unscented Kalman filter (UKF) model called ELM-UKF. This filter can be trained online by the filter residual, innovation and gain matrix of the UKF to simultaneously achieve maneuver feedback and an optimized estimation. Monte Carlo experiments on synthesized radar data showed that our algorithm had better performance on filtering precision compared with most maneuvering target tracking algorithms.

## 1. Introduction

Target tracking is a fundamental and critical task in many sensor-based practical applications including radar-based tracking [1], sonar-based tracking [2], wireless sensor networks [3], video surveillance [4], navigation [5], and mobile robotics [6]. Tracking maneuvering targets is a challenging task because sensor systems are inevitably inaccurate and they are unaware of the uncertain external forces that may be acting on targets, so the target’s dynamic properties cannot be modeled exactly. 

From the viewpoint of statistics, maneuvering targets are often modeled as jump Markov linear systems (JMLS) where the maneuver of the target is modeled as a finite-state Markov chain, and its continuously varying state evolves according to an underlying model that switches among a set of operating models controlled by a Markov chain at each sampling instance [7]. Usually, classical methods for maneuvering target tracking include two main components: (i) maneuver modeling, the stochastic process assumption for an unpredictable maneuver behavior; and (ii) maneuver compensation, the correction of target state estimates to allow for the maneuver. 

For maneuvering target tracking, many algorithms have been developed, and these can be grouped into three types. The first one is based on maneuver dynamics modeling, which is based on the concept of motion-origin uncertainty, and most methods are based on the stochastic process assumption of an unknown acceleration component, such as the Singer model [8], Jerk model [8] and current statistical (CS) model [8], and so on. In the Singer model, target maneuvers are characterized by large deviations from a constant-velocity trajectory corresponding to a random acceleration a(t), which is a zero-mean first-order stationary Markov process with an autocorrelation function. A higher order Jerk model of maneuvering targets has been developed by discretizing a system that contains a third-order derivative of the position. In essence, the CS model is regarded as an improved Singer model with an adaptive mean Markov process. A multitude of practical applications show that the Singer acceleration model is a popular model for target maneuvers. 

Another type of method for maneuvering target tracking is non-linear optimized filtering, which is based on the idea of measurement-origin uncertainty. Many non-linear filtering algorithms are proposed to eliminate the errors caused by nonlinearity in the process of maneuvering target tracking, such as the extended Kalman filter (EKF), unscented Kalman filter (UKF), cubature Kalman filter (CKF) [9] and particle filter (PF) [10], and so on. Among these, the UKF filtering method, compared with the traditional EKF filtering algorithm, does not need to calculate the Jacobian matrix and is more accurate in many tracking scenarios [8,9]. Also, the UKF is more efficient in many practical applications compared with PF filters. 

The third type of algorithm includes learning-based recursive filtering and multi-model based methods, such as multi-layer perceptron (MLP) [11,12,13], the Elman neural network (ENN) [14], fuzzy neural network [15,16], time-delay neural network [17], support-vector regression (SVR) [18,19,20], and interacting multiple model (IMM) filtering [21,22,23,24], and so on. In the mathematical theory of artificial neural networks, the universal approximation theorem states that a feed-forward network with a single hidden layer containing a finite number of neurons can approximate continuous functions on compact subsets of $R^{n}$, under assumptions on the activation function. Specifically, unpredictable and time-variable maneuvers of targets can be modeled by a neural network. For example, a neural-extended Kalman filter (NEKF) proposed by Stubberud [11,12] is a tightly coupled neural network extended Kalman by learning a function that approximates the error between the actual location and the estimated ones. Also, the ENN has been introduced into the IMM filtering model to optimize the IMM algorithm [14]. A standard Kalman filter with a self-constructing neural fuzzy inference network (SONFIN) algorithm [15] for target tracking was developed, where the trained SONFIN can detect when the maneuver occurred, the magnitude of maneuver values, and when the maneuver disappeared. 

In this paper, an ENN-based UKF filter for maneuvering target tracking is proposed, where maneuver parameter feedback and state estimation optimization can be learned simultaneously in the target tracking and filtering framework. 

Our contributions can be summarized as follows. First, a maneuvering target tracking architecture based on ENN and UKF for simultaneous maneuver parameter feedback and state estimation optimization is proposed. Second, the scale factor of process error covariance matrix Q and state vector corrected component is trained by the filter residual, innovation and gain matrix. In addition, the ENN can be supervised and trained online by a sliding-window scheme for different forms of target maneuvers and different sensor measurement noise. 

The remainder of this paper is organized as follows. The description of the basic models and related methods are presented in Section 2. The proposed algorithm is developed in Section 3. Simulation results and performance comparisons are presented in Section 4. Finally, conclusions are provided in Section 5. 

## 2. Basic Models and Related Works

In this paper, radar detection and tracking of air targets is specified as the research scope. As seen in Figure 1, a phased-array radar is tracking maneuvering targets. Phased-array radars have the ability to position quickly and perform adaptive sampling of the target trajectory by flexible beam forming and positioning, however, this relies on a tracking filter that can provide accurate target position prediction, even if the target to be detected is moving at extremely fast speed and in a maneuverable form. In this sense, accurate tracking of maneuvering targets is a core requisite of a high-performance radar. 

Given the radar as the origin of the reference system, the state vector of the maneuvering target at time k is defined as $X_{k}={\left[x,y,z,v_{x},v_{y},v_{z},a_{x},a_{y},a_{z}\right]}^{T}$, the measurement vector as $Z_{k}={\left[r,a,e\right]}^{T}$, where $x,v_{x},a_{x}$, $y,v_{y},a_{y}$ and $z,v_{z},a_{z}$ is the position, velocity and acceleration components of the target in the Cartesian coordinate system, and r,a,e is the range, azimuth and elevation of the target in the Polar coordinate system, respectively. 

Based on the above, a radar system tracking model can be established [1,8,9]:(1)$$\{\begin{array}{c} X_{k}=f\left(X_{k-1}\right)+\Gamma \left(X_{k-1},t_{k}\right)+\omega_{k}\\ Z_{k}=h\left(X_{k}\right)+v_{k} \end{array}$$where f(·) is the target dynamic model, Γ(·) is the dynamic model bias caused by the target maneuver, which is a time-varying nonparametric and unknown component, $\omega_{k}$ is the dynamic model random error with the Gaussian white noise distribution $p\left(\omega_{k}\right)\sim N\left(0,Q_{k}\right)$, h(·) is a non-linear measurement function, and $v_{k}$ is the measurement random error with another Gaussian white noise distribution $p\left(v_{k}\right)\sim N\left(0,R_{k}\right)$. $Q_{k}$ and $R_{k}$ represent the dynamic model error covariance matrix and the measurement error covariance matrix, respectively. 

From Equation (1), we can conclude in short, as follows(2)$$f\left(true\right)=f\left(x_{k}\right)+\Gamma \left(x_{k}\right)$$

The dynamic model bias $\Gamma \left(x_{k}\right)$, also called as maneuver item, can be estimated arbitrarily using a function-approximation scheme that meets the criteria for the Stone-Weierstrauss theorem [11]. 

The conventional methods of dynamic model modeling for maneuvering target tracking mainly include random process assumptions, such as the Markov process and semi-Markov jump process, etc. [8]. One of the most popular models is the Singer model, which defines the $\Gamma \left(X_{k-1},t_{k}\right)+\omega_{k}$ component in Equation (1) as a Markov process and assumes that the target acceleration a(t) is a zero-mean first-order stationary Markov process with an autocorrelation function of $R_{a}\left(\tau \right)=E\left|a\left(t\right)a\left(t+\tau \right)\right|=\sigma^{2}e^{-a\left|\tau \right|}$. Based on these, the dynamic equation of the Singer model would be parameterized by a maneuver factor α [8]. Its variants, called the CS model is essentially a Singer model with an adaptive mean. Another extension of the Singer model is called the Jerk model, which is a derivative of acceleration and a zero-mean high-order stationary Markov process. The Singer model, and its variants, are in essence a kind of a priori model since they do not use online information about the target maneuver, so, they cannot be expected to be very effective for the diverse acceleration situations of actual target maneuvers. 

The IMM algorithm has been widely used in maneuvering target tracking, and it uses the Markov transition probability to switch to multiple models, automatically adjusts the filter bandwidth, and can track arbitrary maneuvering of targets in theory [18]. The transition probability between different models is expressed as a Markov chain. Another approach to unknown covariance matrices and transition models is to rely on multiple standard models and let the filter choose the most adequate one in each time frame, which is effectively modeling it as Jump Markov system [7,8]. Based on the multi-model interactive theory, some models like constant acceleration (CA), constant velocity (CV), constant turning (CT) would be usually used as basic models in an IMM algorithm, and recursive filter algorithms like the Kalman filter (KF), EKF, UKF, etc. are usually applied to iteratively filter for each model, and then the transition probability is used to calculate the current optimal estimate and get the updated covariance matrix. However, the shortcomings of the IMM mean it needs more models and more calculation, moreover, limited models make it difficult to ensure optimal performance. 

Learning-based methods to approximate the target’s unknown maneuver are currently popular. In mathematical theory, the universal approximation theorem states that artificial neural network or support vector regression can approximate any continuous functions. For example, the MLP is tightly coupled with an EKF to form a NEKF filter [11], which learns a function estimate of the biases within the error covariance of the sensor error, and the NEKF can be used to calibrate sensors for the problems of registration and target tracking. Support vector regression (SVR), as another technique for linear and nonlinear function approximation, and is often used for time series prediction [18]. So, an SVR-based Kalman filter for maneuvering target tracking [19,20] also works well. A standard Kalman filter with a self-constructing neural fuzzy inference network (SONFIN) algorithm [15] can be trained to detect when the maneuver occurred, the magnitude of maneuver values, etc. The Elman neural network is also introduced into IMM filtering to form the optimized IMM-ELM algorithm [14]. The Elman neural network is a kind of recurrent neural network with some local memory nodes, which can approximate any time-varying nonlinear function, such as $\Gamma \left(X_{k-1},t_{k}\right)$ in Equation (1). Also, the ENN does not need large training data sets for supervised learning, and the online supervised learning mode enables the neural network to meet the basic requirements of learning. 

The Elman neural network is a typical type of dynamic recurrent neural network (RNN) proposed by Elman in 1990 [25,26]. As shown in Figure 2, the ENN consists of the undertake layer, input layer, hidden layer, and output layer. The undertake layer is used to feedback the output of the hidden layer, along with the signal provided by the input unit at the next time, as the input of the hidden layer unit at the next moment. It can also be considered as a delay operator, so that the network has the function of dynamic memory. Its memory and feedback characteristics make it applicable to adaptive short-term forecasting systems [27,28,29]. In the ENN, $w^{1}$ represents the weight from the context layer to the hidden layer, $w^{2}$ represents the weight from the input layer to the hidden layer, and $w^{3}$ represents the weight from the hidden layer to the output layer, u(k−1) represents the network input vector at the (t−1)th iteration, and y(k) refers to the network output vector at the (t)th iteration. The undertake layer has retained the hidden layer output vector at the previous iteration; that is to say, $x_{c}\left(t\right)$ represents the context layer output vector at the (t)th iteration, and its value equals the hidden layer output vector at the (t−1)th iteration. The thresholds of the hidden layer unit and output layer are $\theta_{j}$ and $\theta_{k}$. 

According to the above configuration, the mathematical model for ENN [25,26] is defined as:(3)$$x\left(k\right)=\lambda \left(w^{1}x_{C}\left(k\right)+w^{2}u\left(k-1\right)\right)$$(4)$$x_{C}\left(k\right)=\alpha x_{C}\left(k-1\right)+x\left(k-1\right)$$
(5)$$y\left(k\right)=g\left(w^{3}x\left(k\right)\right)$$where 0≤α<1 is the self-connected feedback gain factor. λ(x) usually uses the Sigmoid activation function: (6)$$\lambda \left(x\right)=\frac{1}{1+e^{-x}}$$while g(x) is a linear function. In this way, the ENN can approximate any time-varying function with limited discontinuous points at any precision, as long as the neurons in the hidden layer are enough [30]. 

Like most feed-forward neural networks, the ENN also uses the back-propagation (BP) algorithm [31] to train the weights and thresholds. Let the actual output of the system be $y_{d}\left(k\right)$ in step *k*, and its cost function is defined as: (7)$$E\left(k\right)=\frac{1}{2}{\left(y_{d}\left(k\right)-y\left(k\right)\right)}^{T}\left(y_{d}\left(k\right)-y\left(k\right)\right)$$

## 3. Elman Neural Network-Based UKF Filter

### 3.1. Simultaneous Optimization and Feedback Online Learning Scheme

In the classical UKF filter [32,33,34], the dynamic model error covariance matrix $Q_{k}$ and the measurement error covariance matrix $R_{k}$ are always assumed to be time-invariant, but this is not practical in many cases. It is assumed that Ψ(·) is an unscented transform (UT) in matrix form [9], then, according to Equation (1), UKF’s prediction process can be defined as:(8)$$\left({\hat{X}}_{k},{\hat{P}}_{k}\right)=\;\Psi \left(X_{k-1},P_{k-1};f,Q_{k-1},\Theta \right)$$and its update process can be defined as:(9)$$\left({\tilde{X}}_{k},{\tilde{P}}_{k}\right)=\;\Psi \left({\hat{X}}_{k},{\hat{P}}_{k},Z_{k};h,R_{k},\Theta \right)$$where $P_{k-1}$ is a covariance matrix of state $X_{k-1}$, ${\hat{X}}_{k}$ and ${\hat{P}}_{k}$ area predicted state and its corresponding covariance matrix, ${\tilde{X}}_{k}$ and ${\tilde{P}}_{k}$ are the updated state and its corresponding covariance matrix after getting a new measurement $Z_{k}$, and Θ is the parameter set for unscented transform. 

In many references, the recursive filter’s residual [10], innovation [35] and gain matrix [36] can be used to model target maneuvers, and optimized estimation [23,37,38] is also used to track the maneuvering targets. According to the statistical model [8] of maneuvering target tracking, the scale parameter of $Q_{k}$ is a practical parameter to model the maneuver. In our tracking practice analysis, we find that if the scale of the covariance matrix $Q_{k}$ is smaller, then the filtering is more accurate, but when target maneuvering, the scale of $Q_{k}$ should be large enough to ensure that the filter is not be divergent. So, in our update process in Equation (9), some information on the filter’s residual ${\hat{X}}_{k|k-1}-X_{k|k}$, innovation $Z_{k}-h({\hat{X}}_{k|k-1}$) and gain matrix $K_{k}$ are integrated in one neural network to approximate the dynamic model bias to enhance the filter’s performance to adjust the scalar parameter of covariance matrix $Q_{k}$, and to correct the estimated state ${\tilde{X}}_{k}$. 

Based on the above analysis, a simultaneous optimization and feedback online learning scheme on a UKF filter is proposed to learn the dynamic model bias. Specifically, two main factors are tightly coupled in this filter and function simultaneously. One is the feedback strategy, where we use a scale factor $q_{k}$ to adaptively tune the dynamic model error covariance matrix Q and another is the optimization strategy, where we use a corrected component $∆X_{k}$ to refine the final state estimation adaptively.(10)$$\{\begin{array}{l} Q_{k}=q_{k}\cdot Q\\ X_{k}^{\prime }={\tilde{X}}_{k}+∆X_{k} \end{array}$$where $q_{k}$ is the scalar parameter of covariance matrix Q and $∆X_{k}$ is the corrected vector of the estimated state ${\tilde{X}}_{k}$. 

Specifically, an ENN-based learning scheme for the UKF filter [39] was designed in this paper, as seen in Figure 3. In the framework, we define the feedback strategy for the dynamic process error covariance matrix as $Q_{k}$: $Q_{k+1}=q_{k}*Q_{k}$, and we define the optimization strategy for the estimation as ${\tilde{X}}_{k|k}$: $X_{k|k}^{\prime }=∆X_{k|k}+{\tilde{X}}_{k|k}$, where $q_{k}$ is the scale factor of $Q_{k}$, and $∆X_{k|k}$ is the correction component of ${\tilde{X}}_{k|k}$. Ideally, $∆X_{k|k}=X_{k}^{true}-{\tilde{X}}_{k|k}$, and $X_{k}^{true}$ is the real position of the target at time k. 

However, in the practical application, the real location of uncooperative targets is unknown, and the actual dynamic process error covariance $Q_{k}$ is unknown also. We have designed an approximate method; the feedback parameters and corrected state can be learnt online by using a sliding-window scheme. The training of the network is an online supervised training process and the training samples are collected by a sliding window with length N (for example, sample size N = 20). In the initial phase, the ENN is trained by the ground truth, which is available approximately through generating samples like these: $∆X_{k|k}=Z_{k}-{\tilde{X}}_{k|k}$ and $q_{k}=k/N$. In the subsequent process of recursive learning and filtering, $∆X_{k|k}$ and $q_{k}$ can be continuously updated by the network as follows,(11)$$\left(∆X_{k},\;q_{k}\right)=ENN({\hat{X}}_{k|k-1}-{\tilde{X}}_{k|k},\;Z_{k}-h\left({\hat{X}}_{k|k-1}\right),\;K_{k}|\Pi )$$

The learning scheme of recursive filter based on ENN is shown in Figure 3. 

### 3.2. An Elman Neural Network Embedded UKF Filter

Specifically, for a radar tracking a maneuverable target, the 9-dimensional state vector $X_{k|k}={\left[x,v_{x},a_{x},y,v_{y},a_{y},z,v_{z},a_{z}\right]}^{T}$ is defined, and the 3-dimensional measurement vector is $Z_{k}={\left[r,a,e\right]}^{T}$. In the UKF filter, the CA motion model is used to define the dynamic equation of the target, so the dynamic equation f(·) is defined as(12)$$f\left(\cdot \right)=\left[\begin{array}{ccc} ca & 0 & 0\\ 0 & ca & 0\\ 0 & 0 & ca \end{array}\right]where\;ca=\left[\begin{array}{ccc} 1 & T & \frac{T^{2}}{2}\\ 0 & 1 & T\\ 0 & 0 & 1 \end{array}\right]$$while the observation function is coordinate transformation. Initialization of the state covariance matrix P and process noise covariance matrix Q are:(13)$$P_{0}=\left[\begin{array}{ccc} P\_sta & 0 & 0\\ 0 & P\_sta/100 & 0\\ 0 & 0 & P\_sta/1000 \end{array}\right]where\;P\_sta=\left[\begin{array}{ccc} 1.0*e^{1} & 0 & 0\\ 0 & 1.0*e^{1} & 0\\ 0 & 0 & 1.0*e^{1} \end{array}\right]$$(14)$$Q_{0}=\left[\begin{array}{ccc} Q\_sta & 0 & 0\\ 0 & Q\_sta/10 & 0\\ 0 & 0 & Q\_sta/1000 \end{array}\right]where\;Q\_sta=\left[\begin{array}{ccc} 1.0*e^{2} & 0 & 0\\ 0 & 1.0*e^{2} & 0\\ 0 & 0 & 1.0*e^{2} \end{array}\right]$$

The data flow of the detailed filtering algorithm of the proposed Elman Neural Network based UKF Filter (ELM-UKF) is shown in Figure 4. Besides the prediction and update process of classical UKF, there is an ENN-based augmented process for feedback parameter and optimization estimation learning. 

According to Figure 4, the ELM-UKF filtering algorithm can be described in detail as Algorithm 1: 


**Algorithm 1: ELM-UKF Filtering Algorithm**
 1. The UT transform is applied to the filtering value at the last moment, and a set of Sigma point sets and corresponding weights are obtained.(15)$$\chi_{i}^{}=\{\begin{array}{cc} X_{k|k-1} & i=0\\ X_{k-1|k-1}-{\left[\sqrt{\left(n+k\right)\cdot P_{k-1|k-1}}\right]}_{i} & i=1,2,\ldots n\\ X_{k-1|k-1}+{\left[\sqrt{\left(n+k\right)\cdot P_{k-1|k-1}}\right]}_{i} & i=n+1,n+2,\ldots 2n \end{array}$$(16)$$W_{i}^{\left(m\right)}=W_{i}^{\left(c\right)}=\{\begin{array}{cc} \frac{\kappa }{\left(n+\kappa \right)} & i=0\\ \frac{1}{2\left(n+\kappa \right)} & i\neq 0 \end{array}$$ 2. State prediction for Sigma point sets.(17)$$\chi_{i}^{*}=f\left(\chi_{i}\right)$$ 3. The state prediction mean and covariance matrix is calculated by the weighted Sigma point.(18)$${\hat{X}}_{k|k-1}^{*}=\sum_{i=0}^{2n}W_{i}^{\left(m\right)}\chi_{i}^{*}\;$$(19)$$P_{k|k-1}=\sum_{i=0}^{2n}W_{i}^{\left(c\right)}\left(\chi_{i}^{*}-{\hat{X}}_{k|k-1}\right){\left(\chi_{i}^{*}-{\hat{X}}_{k|k-1}\right)}^{T}+Q_{k}$$ 4. A new Sigma point set is obtained by performing UT transformation on the state prediction.(20)$$\chi_{i}^{\left(z\right)}=\{\begin{array}{cc} {\hat{X}}_{k|k-1} & i=0\\ {\hat{X}}_{k|k-1}-{\left[\sqrt{\left(n+k\right)\cdot P_{k|k-1}}\right]}_{i} & i=1,2,\ldots n\\ {\hat{X}}_{k|k-1}+{\left[\sqrt{\left(n+k\right)\cdot P_{k|k-1}}\right]}_{i} & i=n+1,n+2,\ldots 2n \end{array}$$ 5. Observation prediction for the new Sigma point sets.(21)$$z_{i}^{*}=h\left(\chi_{i}^{\left(z\right)}\right)\;,i=1,\ldots ,2n$$ 6. The observation prediction, observation covariance and the innovation covariance matrix are calculated by the weighted predictive value.(22)$$\{\begin{array}{l} {\hat{Z}}_{k|k}=\sum_{i=0}^{2n}W_{i}^{\left(m\right)}z_{i}^{*}\\ P_{xz}=\sum_{i=0}^{2n}W_{i}^{\left(c\right)}\left(\chi_{i}^{*}-{\hat{X}}_{k|k-1}\right)\cdot {\left(z_{i}^{*}-{\hat{Z}}_{k|k}\right)}^{T}\\ P_{zz}=\sum_{i=0}^{2n}W_{i}^{\left(c\right)}\left(z_{i}^{*}-{\hat{Z}}_{k|k-1}\right)\cdot {\left(z_{i}^{*}-{\hat{Z}}_{k|k}\right)}^{T}+R_{k} \end{array}$$ 7. Filter gain matrix.(23)$$K_{k}=P_{xz}P_{zz}^{-1}$$ 8. Update the current state and state covariance matrix.(24)$${\tilde{X}}_{k|k}={\hat{X}}_{k|k-1}+K_{k}\left(Z_{k|k}-{\hat{z}}_{k|k}\right)\;$$(25)$$P_{k|k}=P_{k|k-1}-K_{k}P_{zz}K_{k}^{T}$$ 9. Calculate the residual, innovation and gain matrix, and they are used as the input vector. When training, the neural function ENN (∙) is trained and supervised b Algorithm 2 and the network parameter Π can be learnt when Algorithm 2 is converged. When filtering, the corrected estimate item and the scale factor of *Q* matrix can be obtained by the forward steps of ENN.(26)$$\left(∆X_{k|k},\;q_{k}\right)=ENN({\hat{X}}_{k|k-1}-{\tilde{X}}_{k|k},\;Z_{k}-h\left({\hat{X}}_{k|k-1}\right),\;K_{k}|\Pi )$$ 10. Update the process noise covariance matrix and get the corrected estimate.(27)$$Q_{k}=q_{k}\cdot Q$$(28)$$X_{k|k}^{\prime }={\tilde{X}}_{k|k}+∆X_{k|k}$$

Before training, historical tested data based on a standard UKF filter are constructed as training samples, the input vector $u\left(k\right)=({\hat{X}}_{k|k-1}-{\tilde{X}}_{k|k},\;Z_{k}-h\left({\hat{X}}_{k|k-1}\right),\;K_{k})$ and the output vector $\hat{y}\left(k\right)=\left(∆X_{k|k},\;q_{k}\right)$ are collected. Then, the main part of the Elman neural network training algorithm is described as Algorithm 2: 


**Algorithm 2: ENN Training Algorithm**
 1. Network parameter initialization: create a case of an Elman neural network with 9 input units, 18 hidden units and 4 output units. Initialize the weights and learning rate, $w^{1},w^{2},w^{3},l$. 2. Get the input vector $u\left(k\right)=({\hat{X}}_{k|k-1}-{\tilde{X}}_{k|k},\;Z_{k}-h\left({\hat{X}}_{k|k-1}\right),\;K_{k})$ and the output vector $\hat{y}\left(k\right)=\left(∆X_{k|k},\;q_{k}\right)$ for training. 3. The input layer data and the undertake layer data are weighted, and added as the input of the hidden layer.(29)$$\hat{x}\left(k\right)=w^{1}x_{c}\left(k\right)+w^{2}u\left(k\right)$$ 4. Use the hidden layer’s Sigmoid activation function to obtain the output of the hidden layer as the input of the output and undertake layer.(30)$$x_{c}\left(k\right)=x\left(k\right)=\lambda \left(\hat{x}\left(k\right)\right)$$ 5. The output layer is the weighted linear combination of the input, and obtains the output value.(31)$$y\left(k\right)=g\left(w^{3}x\left(k\right)\right)$$ 6. Calculate the error based on following loss function. If the loss is less than a threshold ε (ε > 0), then end the training. (32)$$E\left(k\right)=\frac{1}{2}{\left(y\left(k\right)-\hat{y}\left(k\right)\right)}^{T}\left(y\left(k\right)-\hat{y}\left(k\right)\right)$$ 7. Calculate the weight update according to the BP algorithm. (33)$$w^{1}=w^{1}+l\cdot \left(y\left(k\right)-\hat{y}\left(k\right)\right)\cdot w^{3}\cdot x\left(k\right)$$(34)$$w^{2}=w^{2}+l\cdot \left(y\left(k\right)-\hat{y}\left(k\right)\right)\cdot w^{3}\cdot u\left(k\right)$$(35)$$w^{3}=w^{3}+l\cdot \left(y\left(k\right)-\hat{y}\left(k\right)\right)\cdot x\left(k\right)$$

## 4. Experiments and Discussion

Simulation experiments were designed for a variety of different maneuvering forms, such as double S-shaped, single S-shaped and helical trajectories; different random errors were also superimposed on each maneuvering form. In our experiments, the simulated target flight trajectories are based on the combination of five basic motion equations, such as line, parabola, serpentine and hyperbola and a joint between two consequent segments, at each different trajectory segment or joint, where the acceleration is variable or with a different configuration. We used a trajectory edit toolkit to generate different flights trajectories with or without maneuver. According to the design of an uncooperative target tracking, there is inevitable, unknown and time-varying dynamic model bias between the real trajectory and the filter’s motion equations, which is the maneuver item. 

After generating the trajectories, then the position of the target was transformed to a radar-centered reference system and was superimposed with specific random distribution error to get the measurement series. In most physical radar systems, the random component of measurement error after systematic calibration can be approximated to be a Gaussian distribution. 

The 1000 Monte Carlo experiments were carried out to compare the performance of classical and the state-of-the-art models like the Singer [8], a CA model and CV model combined, simplified IMM-KF [9], IMM-ELM [14], SVR-UKF [19,20], etc., and our proposed ELM-UKF. In each group of comparison experiments, the motion equations and initial parameters of these covariance matrixes P, Q and R were set with the same parameters and the CA model was selected in these methods. By qualitative and quantitative analyses of the convergence and filtering precision of the above five models, it was found that the online learning model based on the Elman neural network is superior to the state-of-the-art methods. 

### 4.1. Experiments on Algorithm Convergence

In the experimental scenario, the target is flying at the initial speed of v=200 m/s and the constant acceleration of $a=2\;m/s^{2}$ in a single S-shaped trajectory, and the maximum maneuvering acceleration is $a_{max}=64\;m/s^{2}$, which is a general maneuvering scenario. The Gaussian white noise of measurement error with range component ∆R=300 m, azimuth component ∆A=0.5° and elevation component ∆E=0.5° is added to the trajectory in the scenario, as in Figure 5a. Figure 5a also shows the superimposed trajectories and their error distribution for the different filtering models, which are the Singer, IMM-KF, IMM-ELM, and SVR-UKF models and our proposed ELM-UKF model in a maneuver scenario. The filtering estimation error distributions for the different methods are shown in Figure 5b–d according to range, azimuth and elevation, respectively. 

Additionally, the statistical results of classical and the state-of-the-art models are listed in Table 1. In Table 1, the metric is calculated as $\mathrm{Mean}\_X=\overset{N}{\underset{i=1}{\sum }}\left({\hat{X}}_{i}-X_{i}\right)/N$, where $\hat{X}$ is the filter estimation, X is the ground truth, and X can be range (*R*), azimuth (*A*) or elevation (*E*) separately, and N is the total number of measurements of this scenario. $\mathrm{RMSE}\_X={\left(\overset{N}{\underset{i=1}{\sum }}{\left({\hat{X}}_{i}-\mathrm{Mean}\_X\right)}^{2}/\left(n-1\right)\right)}^{1/2}$, which is the root mean of square error (RMSE) of the range and azimuth and elevation. 

Based on qualitative and quantitative analysis of the experiment results of maneuvering target tracking, it was found that these four filtering models can achieve good convergence. Generally, they converge within 20~50 measurements from initial filtering using classical methods. Moreover, the online learning method based on the Elman neural network has the fastest convergence performance among the five models from the range error distribution, and it is convergent within 10 measurements.

### 4.2. Experiments on Different Maneuvering Forms

In scenario 1, the target is flying at a constant speed of v=200 m/s in a double S-shaped trajectory, and the maximum maneuvering acceleration is $a_{max}=16\;m/s^{2}$, which is a weak maneuvering form. In scenario 2, the target is flying at the initial speed of v=200 m/s and the constant acceleration of $a=2\;m/s^{2}$ in a single S-shaped trajectory, and the maximum maneuvering acceleration is $a_{max}=64\;m/s^{2}$, which is a general maneuvering form. In scenario 3, the target is flying at a constant speed of v=300 m/s in a helical trajectory, and the maximum maneuvering acceleration is $a_{max}=90\;m/s^{2}$, which is a strong maneuvering form. The Gaussian white noise of measurement error with ∆R=300 m,∆A=0.5°,∆E=0.5° is added to the trajectories in the above three scenarios. Figure 6a–c are three kinds of maneuver scenarios which are superimposed by different filtered trajectories, for example, the Singer, IMM-KF, IMM-ELM, and SVR-UKF models and our proposed model. The distribution of mean and RMSE of 1000 Monte Carlo simulations for the three maneuver scenarios are shown in Figure 7a–c. The statistical results of filtering precision are shown in Table 2. 

In Figure 7 and Table 2, the position estimation $\;\hat{p}$ means the target’s state estimation weighted distance in a Cartesian coordinate system, which is defined as follows: $\;\hat{p}=\sqrt{{\hat{x}}^{2}+{\hat{y}}^{2}+{\hat{z}}^{2}}$, and the position error is the difference between the position estimation $\;\hat{p}$ and the position of ground truth, and the error mean and RMSE are the corresponding statistical mean and root mean square error.

From Figure 7, it can be seen that all five methods, including our proposed algorithm can effectively adapt to the above three maneuvering scenarios and it can be concluded that when the target is moving in maneuvers, the filter precision of the traditional Singer model decreases. Compared with the Singer model, the IMM-KF model, SVR-UKF model and IMM-ELM model have stronger immunity to the target’s maneuverability, and the filtering precision was improved a little. Either from the statistical mean or RMSE of Monte Carlo experiments, it can be seen that our proposed ELM-UKF filter has the best filtering precision among the five models. We used the ratio of the statistical mean of the filtered error to measurement error to quantitatively evaluate the performance. From Table 2, it can be concluded that our proposed model further reduced the ratio by 16.35% on average and up to 24.39%, and the ratio of RMSE of our proposed filter is further reduced 20.26% on average and up to 27.67% compared with other four models.

### 4.3. Experiments on Different Levels of Measurement Error

In order to assess the influence of different levels of sensor measurement error on filtering when the target is moving in a maneuvering form, three levels of Gaussian white noise measurement error were superimposed on the real trajectory. For instance, the largest measurement error was set as ∆R=500 m, ∆A=1.0°, ∆E=1.0°, and the middle level was set as ∆R=300 m, ∆A=0.5°,  ∆E=0.5° and the smallest level was set as ∆R=50 m,∆A=0.2°,∆E=0.2°. After adding different-level measurement errors, the filtered trajectories of the Singer, IMM-KF, IMM-ELM, and SVR-UKF models and our proposed ELM-UKF model were superimposed as shown in Figure 8a–c. The statistical mean and RMSE of 1000 Monte Carlo simulations of the scenarios added by three levels of Gaussian random measurement error are shown in Figure 9a–c. The statistical results of the filtering precision are shown in Table 3.

From the Monte Carlo experiments, it was found that our proposed model can effectively adapt to different levels of measurement error. Also, it could be concluded that the filtering precision of our proposed method is superior to the other four methods, including the Singer model, IMM-KF model, SVR-UKF model and IMM-ELM model under different levels of measurement error, and that the precision improvement ratio was more than 20%. 

Moreover, compared with the other four models, the ratio of the statistical mean of the filtered error of our algorithm was further reduced by 18.19% on average and up to 24.83%, and the corresponding statistical RMSE of the filtered error was further reduced by 20.16% on average and up to 27.91% above the noise levels of the middle random error. Even on the lower noise level, the ratio of the statistical mean and statistical RMSE of our proposed filter was further reduced by an average of 12.05% and 12.64%, respectively.

## 5. Conclusions

In this paper, a novel method for maneuvering target tracking learnt by the Elman neural network is proposed. It is based on a simultaneous optimization and feedback learning scheme to optimize the filtering performance. The ENN algorithm is embedded into a classical UKF to learn to track maneuvering targets. The ENN is trained online by the UKF filter’s residual, innovation and gain matrix as input and it can learn to tune a scale factor of the dynamic model error covariance matrix and to correct the final state estimation simultaneously, even if the sensor is tracking an uncooperative maneuvering target. It can be concluded after numerous of Monte Carlo experiments, that our proposed method is superior to the state-of-the-art methods, like the Singer model, IMM-KF model, SVR-UKF model and IMM-ELM model on different maneuvering forms and different measurement errors. 

## Figures and Tables

**Figure 1 sensors-19-01596-f001:**
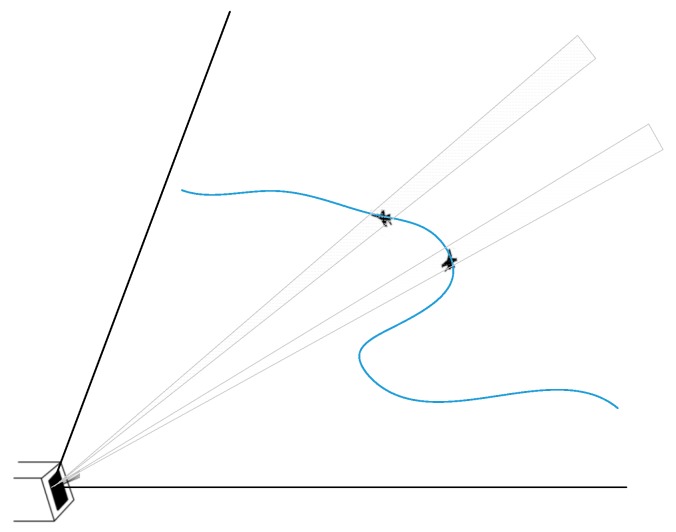
Radar tracking maneuvering targets.

**Figure 2 sensors-19-01596-f002:**
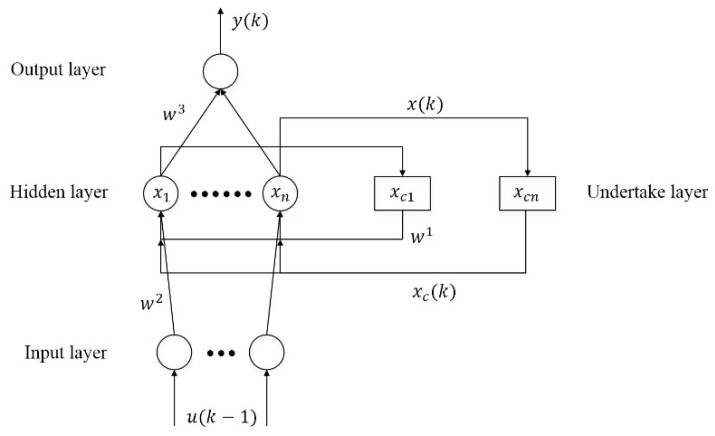
The Elman neural network.

**Figure 3 sensors-19-01596-f003:**
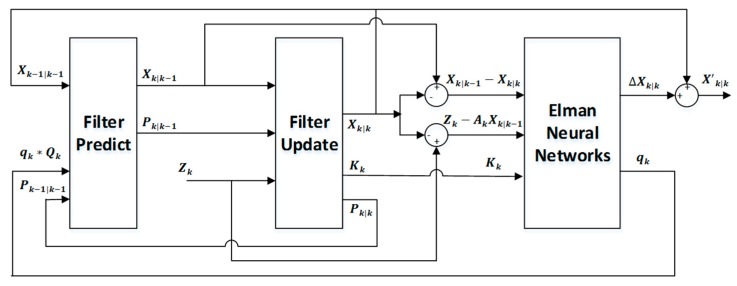
A learning scheme of recursive filter based on the Elman neural network.

**Figure 4 sensors-19-01596-f004:**
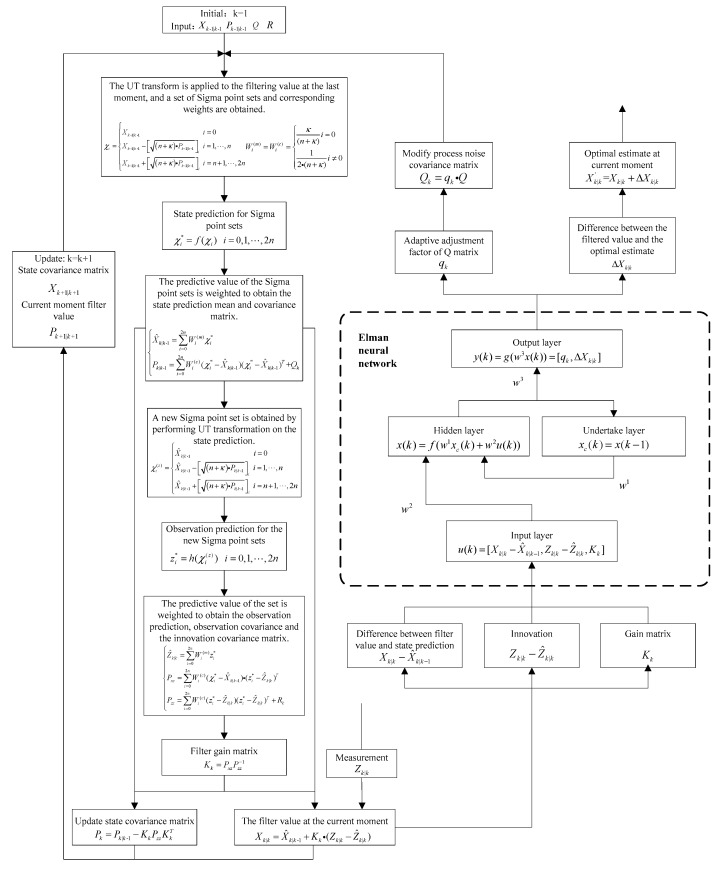
ELM-UKF filtering algorithm flow.

**Figure 5 sensors-19-01596-f005:**
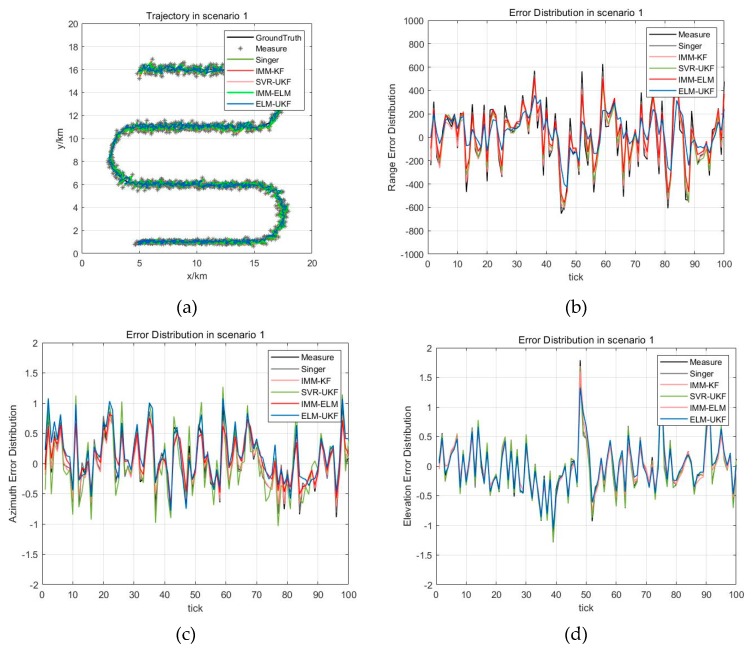
Trajectory and estimation error distribution for scenario 1. (**a**) Trajectory for scenario 1; (**b**) Range estimation error distribution of different methods; (**c**) Azimuth estimation error distribution of different methods; (**d**) Elevation estimation error distribution of different methods.

**Figure 6 sensors-19-01596-f006:**
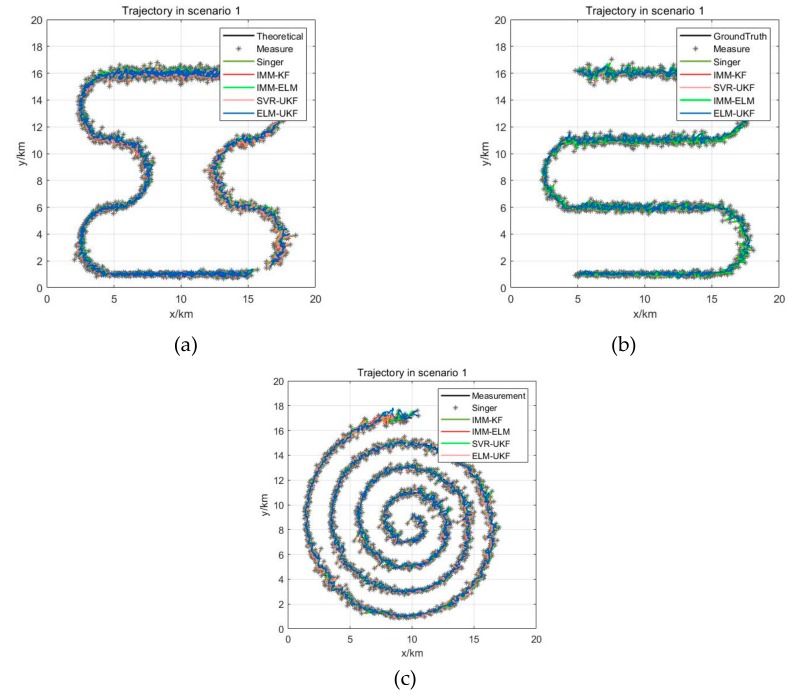
Different maneuver scenarios. (**a**) Superimposed trajectory of weak maneuvering form as scenario 1; (**b**) Superimposed trajectory of general maneuvering form as scenario 2; (**c**) Superimposed trajectory of strong maneuvering form as scenario 3.

**Figure 7 sensors-19-01596-f007:**
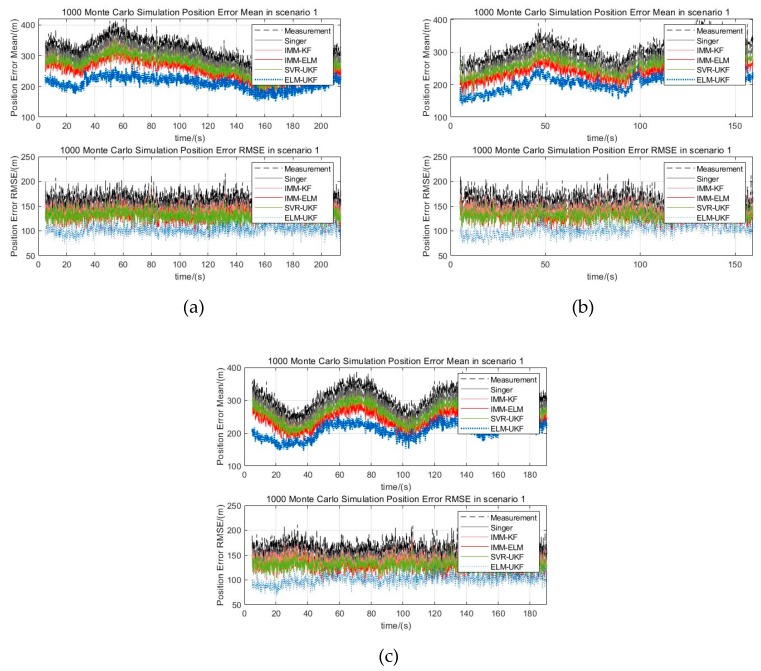
Statistical mean and RMSE of Monte Carlo experiments for different scenarios. (**a**) Position error of mean and RMSE distribution for scenario 1; (**b**) Position error of mean and RMSE distribution for scenario 2; (**c**) Position error of mean and RMSE distribution for scenario 3.

**Figure 8 sensors-19-01596-f008:**
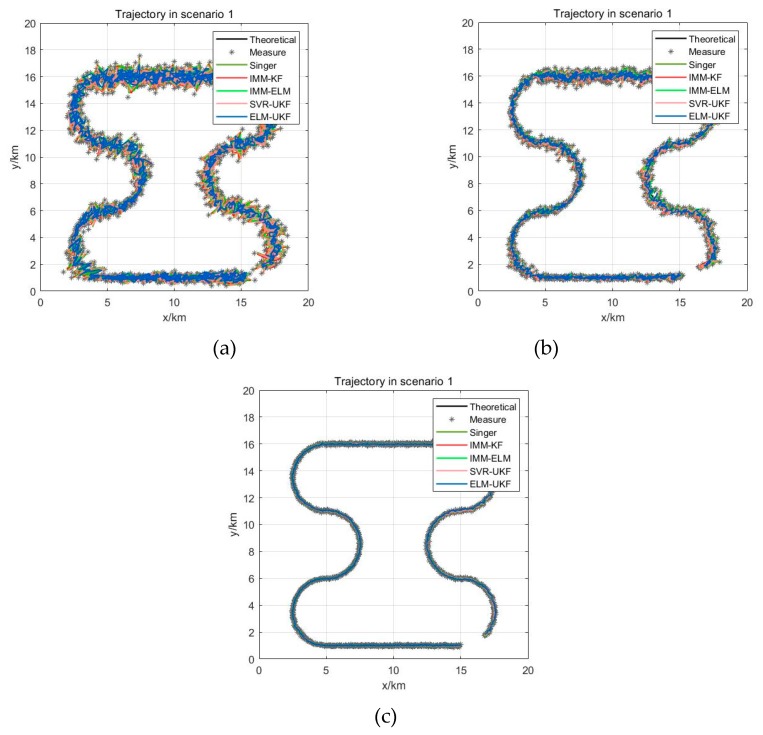
Superimposed trajectories with different filters and different levels of measurement error. (**a**) With large random error; (**b**) With medium random error; (**c**) With small random error.

**Figure 9 sensors-19-01596-f009:**
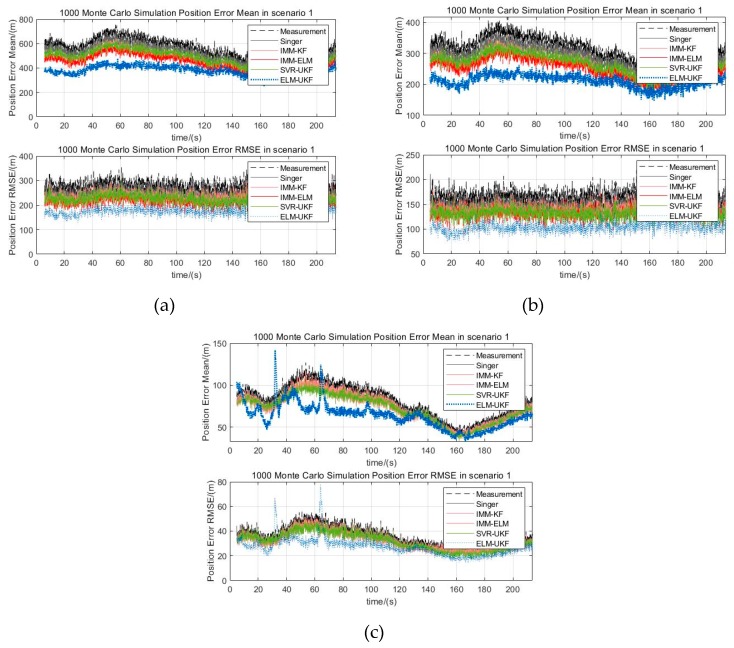
Statistical mean and RMSE of Monte Carlo experiments under three levels of random error. (**a**) Statistical mean and RMSE with large random error; (**b**) Statistical mean and RMSE with medium random error; (**c**) Statistical mean and RMSE with small random error.

**Table 1 sensors-19-01596-t001:** Statistical mean and RMSE of filtering estimation error with different methods.

	Mean_R (m)	Mean_A (deg)	Mean_E (deg)	RMSE_R (m)	RMSE_A (deg)	RMSE_E (deg)
Measurement	−4.1979	−0.0117	−0.0219	303.0576	0.4780	0.4887
Singer	−1.0779	−0.0122	−0.0216	270.1333	0.4601	0.4747
IMM-KF	−3.4715	−0.0110	−0.0222	253.9812	0.4027	0.4283
IMM-ELM	+18.3676	−0.0074	−0.0032	232.0766	0.3674	0.4058
SVR-UKF	−1.6511	−0.0129	−0.0245	240.2092	0.4670	0.4690
ELM-UKF	+32.4549	+0.0187	+0.0041	180.8674	0.3968	0.3896

**Table 2 sensors-19-01596-t002:** Monte Carlo simulation error statistics for three scenarios.

Models	Scenario 1		Scenario 2		Scenario 3	
	Position Error Mean (m)	Position Error RMSE (m)	Position Error Mean (m)	Position Error RMSE (m)	Position Error Mean (m)	Position Error RMSE (m)
Measurement	312.0497	167.7742	307.2045	166.2080	302.6656	165.0034
Singer	286.0601	147.8417	284.0687	148.0820	276.9512	145.0397
IMM-KF	265.2177	137.9686	263.0705	137.9041	256.3642	136.8777
IMM-ELM	210.6755	130.6300	244.6634	128.1961	257.3732	130.0383
SVR-UKF	264.8515	131.5167	263.7172	131.2646	238.4987	127.3800
ELM-UKF	209.9392	103.2325	209.9474	102.0898	207.3112	101.7506

**Table 3 sensors-19-01596-t003:** Statistical mean and RMSE of Monte Carlo simulation under different random errors.

Models	Large Random Error		Middle Random Error		Small Random Error	
	Position Error Mean (m)	Position Error RMSE (m)	Position Error Mean (m)	Position Error RMSE (m)	Position Error Mean (m)	Position Error RMSE (m)
Measurement	556.2067	285.7184	312.0497	167.7742	80.7968	40.7545
Singer	513.3084	252.3426	286.3558	150.5387	75.5647	33.1444
IMM-KF	475.2431	232.6749	265.4042	138.6168	74.4837	32.8079
IMM-ELM	476.9646	223.7813	250.6126	131.2331	70.6471	31.2148
SVR-UKF	442.5600	216.4022	264.9855	132.0531	72.0840	31.5407
ELM-UKF	375.1780	174.7110	210.4447	103.7043	63.4614	27.0275

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
