# Peer review of "Maneuvering Target Tracking Using Simultaneous Optimization and Feedback Learning Algorithm Based on Elman Neural Network"

_sensors, 2019, doi:10.3390/s19071596_

Round 1
Reviewer 1 Report
This paper proposed a simultaneous optimization and feedback learning algorithm based on Elman neural network (ENN) for maneuvering target tracking. The topic sounds interesting. However, in order to further improve this paper’s quality, some minor suggestions for the authors are provided as follows: There are some typos in the paper. The authors should be check again. Line 257, “Figure 4(a)” should be “Figure 5(a)”. The results sound interesting in the paper, but the authors may state clearly in simulation assumptions. More models and simulation results are needed to show your proposed algorithms’ performance. Why used the parameter setting of the Gaussian white noise of measurement error with delta_Range (R), delta_Azimuth (A) and delat_Elevation (E) , authors may state clearly. The authors may also state clearly their proposed methods and contributions of the paper. In general, I would recommend to be accepted the paper for publication after minor revision.
Author Response
Response to Reviewer 1 Comments
Point 1: The authors should be check again. Line 257, “Figure 4(a)” should be “Figure 5(a)”.
Response 1: Yes, line 296, “Figure 4(a)” has been corrected to “Figure 5(a)”.
Point 2: The results sound interesting in the paper, but the authors may state clearly in simulation assumptions.
Response 2: We have added more description on simulation assumptions in Sec. 4 (Experiments and Discussion), one is on the target’s trajectories generation, and another is on the sensor measurement error assumption.
The added description on target’s trajectories generation is on the Line 274 to Line 281 as follows, “In our experiments, the simulated target flight trajectories are based on the combination of five basic motion equations, such as line, parabola, serpentine and hyperbola and a joint between them, at each trajectory segment or joint, where its acceleration is variable or with different configuration. Ours trajectory edit toolkit can be used to generate any flights trajectories with maneuver. After generated the trajectories, then the position is transformed to radar-centered reference and superposed with specific random distribution error to get the measurement series.”
The statement on sensor measurement error assumption is introduced in different experiment cases separately. For example, the description of sensor measurement is on the Line 294 to Line 296 stated as “The Gaussian white noise of measurement error with range component ∆R=300m, azimuth component ∆A=0.5° and elevation component ∆E=0.5° is superposed to the trajectory in the scenario as Figure 5(a).” In the Section 4.2 (Line 409 to Line 411), the sensor measurement error assumption is stated as “The Gaussian white noise of measurement error with ∆R=300m,∆A=0.5°,∆E=0.5° is added to the trajectories in the above three scenarios”. In the section 4.3 (Line 450 to Line 452), the sensor measurement error assumption is stated as “For instance, the large measurement error is set ∆R=500m, ∆A=1.0°, ∆E=1.0°, and the middle measurement error is set ∆R=300m, ∆A=0.5°, ∆E=0.5°and the small measurement error is set ∆R=50m,∆A=0.2°,∆E=0.2°.”
Point 3: More models and simulation results are needed to show your proposed algorithms’ performance.
Response 3: Actually, the models and algorithm for maneuvering target tracking are emerging in these years, moreover many summaries in the several survey papers from Pro. X. R. Li. From thousands of papers on maneuvering target tracking, it can be divided into about 4 categories according to the surveys of Pro. X. R. Li, and in each category, we select a classical algorithm to compare. For example, Singer model is the classical method of dynamic modeling based on statistics, the UKF is a classical method based on Non-linear filtering, and IMM model is the classical method based multiple model. And machine learning is a kind of new method, we select a comparable and classical method ELM_IMM. According to your advice, we have added another comparable machine learning based method, which is the support vector regression (SVR) based maneuvering target tracking in the revised version.
The SVR method is added as “Support Vector Regression (SVR), as another techniques for linear and nonlinear function approximations, often is used for time series prediction. So SVR based Kalman filter for maneuvering target tracking [37, 38, 39] also works well.” in the Line 152 to Line 154, and the added experiments are in the Fig. 5 and Table 1.
P. S.: surveys of Pro. X. R. Li on maneuvering target tracking listed as follows
[1] X. Rong Li, Vesselin P. Jilkov, Survey of Maneuvering Target Tracking. Part I: Dynamic Models, IEEE Transactions on Aerospace and Electronic Systems, 39(4):1333–1364, 2003.
[2] X. Rong Li, Vesselin P. Jilkov. Survey of Maneuvering Target Tracking. Part II: Motion Models of Ballistic and Space Targets. IEEE Transactions on Aerospace and Electronic Systems, 46(1): 96–119, 2010.
[3] X. Rong Li, Vesselin P. Jilkov. Survey of maneuvering target tracking: decision-based methods. Proceedings Volume 4728, Signal and Data Processing of Small Targets, 2002.
[4] Li, Xiao-Rong, Jilkov Vesselin P., A survey of maneuvering target tracking: approximation techniques for nonlinear filtering, Proceedings of the SPIE, Volume 5428, p. 537-550, 2004.
[5] X. Rong Li, V.P. Jilkov, Survey of maneuvering target tracking. Part V. Multiple-model methods, IEEE Transactions on Aerospace and Electronic Systems, 41(4): 1255-1321, 2005.
Point 4: Why used the parameter setting of the Gaussian white noise of measurement error with delta_Range (R), delta_Azimuth (A) and delat_Elevation (E) , authors may state clearly.
Response 4: In order to verify the effectiveness of different models and algorithms, noise with Gaussian distribution assumption is usually used in state-space model and Kalman-like filters. On one hand, from optimal estimation theory, the conclusion of minimum root mean square error (RMSE) of recursive filter is based on the assumption of measurement error with Gaussian white noise distribution, which is mentioned in most famous monographies and many papers. On the other hand, in most physical radar systems, the random component of measurement error after systematic calibration can be approximated to be a Gaussian distribution.
To state more clearly, we added some explanation in the Line 280 to Line 283 as “After generating the trajectories, then the position of target is transformed to radar-centered reference and superimposed with specific random distribution error to get the measurement series. Moreover, in most physical radar systems, the random component of measurement error after systematic calibration can be approximated to be a Gaussian distribution.”
Point 5: The authors may also state clearly their proposed methods and contributions of the paper.
Response 5: The contribution summary of the proposed methods is stated in the Line 74 to Line 78 in the initial version. After this revision, we have changed as “First, a maneuvering targets tracking architecture based on ENN and UKF for simultaneous maneuver parameter feedback and state estimation optimization is proposed. Second, the scale factor of Q and state corrected component is trained by the filter residual, innovation and gain matrix. In addition, the ENN can be supervised trained online by a sliding-window scheme for different target maneuver form and different sensor measurement noise.” in the Line 75 to Line 80.

Reviewer 2 Report
In this paper, a simultaneous optimization and feedback learning algorithm for maneuvering target tracking based on Elman neural network (ENN) is proposed. These two strategies are integrated in an ENN based Unscented Kalman Filter (UKF) called ELM-UKF. And this filter can be online trained by the filter residual, innovation and gain matrix of UKF to achieve a maneuver feedback and an optimized estimation simultaneously.
The detailed comments are as follows:
1. Could you further clarify the contribution of this paper, especially point out the difference from reference [36]?
2. Eq.(10) shows that the Q is corrected through a scale parameter q_{k}. In fact, the different dimensions of Q represent the modeling uncertainty of different states. Then I think that the Eq. (10) may be only a special case and does not have universal applicability. Is it possible to make the algorithm more widely used through adding more parameters in Eq. (10)?
3. As we see from Figure 4, the Q_{k} corrected at time k is directly used for the filtering calculation at time k+1, is it possible to iterate the UKF and ENN loops at time k to continuously correct Q_{k} and X_{k} to further improve the estimation accuracy?
4. In the simulation experiments, how to choose/design the values of the parameters? For example, the parameter Q and the model set in IMM-KF and IMM-ELM should be given.
5. Some abbreviations are not familiar to the readers. For clarity, please give their full names/forms when they first came out.
6. The multiple-model methods are the mainstream technique for maneuvering target tracking, and there are some new results on this topic recent years which are as follows for your information
[1] Route-based dynamics modeling and tracking with application to air traffic surveillance. IEEE TITS, DOI: 10.1109/TITS.2018.2890570.
[2] Hybrid grid multiple-model estimation with application to maneuvering target tracking. IEEE TAES, 2016.
Author Response
Response to Reviewer 2 Comments
Point 1: Could you further clarify the contribution of this paper, especially point out the difference from reference [36]?
Response 1: Reference [36] is our earlier results on maneuvering target tracking published on ACPR 2017.
On the basis of reference [36], firstly, we investigated the formal model of Elman Neural Network-based UKF learning algorithm in section 3.1 more deeply and provided more detailed description about algorithm dataflow in section 3.2.
Secondly, the learning method is updated to a sliding-window scheme, which means the tracking and learning can be done simultaneously. The description is stated as “We have designed an approximate method, the feedback parameter and corrected state can be learnt online by a sliding-window scheme. The training of the network is an online supervised training process and the training samples are collected by a sliding window with length N (for example, sample size N=20). In the initial phase, the ENN is trained by the ground truth, which is available approximately through generating samples like these: ∆X(k|k)=Z_k-X(k|k) and qk=k/N. In the subsequent process of recursive learning and filtering, ∆X(k|k) and qk can be continuously updated by the network” in the Line 228 to Line 236.
Moreover, we add some experiments of other comparable methods, like SVR based UKF method in the revised version. The added content is in the Fig. 5 and Table 1.
At last, the writing has been improved further in this manuscript.
Point 2: Eq.(10) shows that the Q is corrected through a scale parameter q_{k}. In fact, the different dimensions of Q represent the modeling uncertainty of different states. Then I think that the Eq. (10) may be only a special case and does not have universal applicability. Is it possible to make the algorithm more widely used through adding more parameters in Eq. (10)?
Response 2: As we defined in the paper, qk is a scale factor of Q. According to the definition of Q, if the state vector , then covariance matrix Q will be a matrix , which has elements. According to the statistical model of maneuvering target tracking, the scale parameter qk of Q is an idea parameter to model the maneuver. And in our analysis of tracking practice, the scale of the Q is smaller, then the filtering is more accurate, but when target maneuvering, if scale of Q should be larger to make sure the filter would not be divergent. So, when tracking maneuvering targets, the scale of Q is an idea parameter should be tuned dynamically. And the scale parameter can be learnable in a feedback form in our method.
To state more clearly, we added some explanation in the Line 204 to Line 207 as “According to the statistical model [8] of maneuvering target tracking, the scale parameter of Qk is an idea parameter to model the maneuver. And in our analysis of tracking practice, the scale of the Qk is smaller, then the filtering is more accurate, but when target maneuvering, if scale of Qk should be larger to make sure the filter would not be divergent.”
Point 3: As we see from Figure 4, the Q_{k} corrected at time k is directly used for the filtering calculation at time k+1, is it possible to iterate the UKF and ENN loops at time k to continuously correct Q_{k} and X_{k} to further improve the estimation accuracy?
Response 3: In the mathematical model of optimal filtering and state space model, the filtering process is modeled as a Markov Process, which means the current state only relies on the state of the last time, and the original idea of Kalman filter is to construct a recursive estimator with minimum Root Mean Square Error (RMSE). So in Kalman filter, recursive scheme is used in the state space model and optimal filtering. Iterating the UKF and ENN loops is an amazing idea, after analysis we find it cannot guarantee steady and would lead the filter to be divergent and to be delayed. Maybe it would work better in some special cases. But from a general point of view, we should carefully choose this iterative scheme.
Point 4: In the simulation experiments, how to choose/design the values of the parameters? For example, the parameter Q and the model set in IMM-KF and IMM-ELM should be given.
Response 4: Yes, in order to indicate the comparison is fair, the motion equations and initial parameters of process noise covariance, state error covariance, and measurement noise covariance are set to the same. In our paper, we select CA model in most methods. And initial parameter of P, Q and R of different filtering methods are also the same.
To state more clearly, we added some explanation in the Line 286 to Line 287 as “In the comparison experiments, the motion equations and initial parameters of these covariance matrixes P, Q and R are set with the same parameters. And CA model is selected in these methods.”
Point 5: Some abbreviations are not familiar to the readers. For clarity, please give their full names/forms when they first came out.
Response 5: We have checked all abbreviations and added the full name of some abbreviations when it appears first, such as KF, EKF, CKF, PF, NEKF, CA etc.
Point 6: The multiple-model methods are the mainstream technique for maneuvering target tracking, and there are some new results on this topic recent years which are as follows for your information.
[1] Route-based dynamics modeling and tracking with application to air traffic surveillance. IEEE TITS, DOI: 10.1109/TITS.2018.2890570.
[2] Hybrid grid multiple-model estimation with application to maneuvering target tracking. IEEE TAES, 2016.
Response 6: The models and algorithm for maneuvering target tracking are emerging in these years, moreover many summaries in the several survey papers from Pro. X. R. Li. From thousands of papers on maneuvering target tracking, it can be divided into 3-4 categories according to the surveys of Pro. X. R. Li in general, and in each categories, we select a classical and representative one as our comparing method such as Singer, UKF, IMM, ELM_IMM, SVR_UKF etc. As you said, multi-model method is one of the popular and important methods for maneuvering target tracking. In the section of introduction and related works, we have mentioned the multi-model method in the Line 58 to Line 60 and Line 133 to Line 146. In this paper, our proposed Elman Neural Network based feedback and optimization learning method is another novel idea to try to resolve this problem, they are comparable at the similar target trajectories and sensor error assumption. Actually, we have tested the tracking performance of ELM_IMM and IMM, which is similar as the reference you recommended, and compared them with our proposed method. The results can be seen from Figure 5 to Figure 9 and from Table 1 to Table 3.
P. S.: surveys of Pro. X. R. Li on maneuvering target tracking listed as follows
[1] X. Rong Li, Vesselin P. Jilkov, Survey of Maneuvering Target Tracking. Part I: Dynamic Models, IEEE Transactions on Aerospace and Electronic Systems, 39(4):1333–1364, 2003.
[2] X. Rong Li, Vesselin P. Jilkov. Survey of Maneuvering Target Tracking. Part II: Motion Models of Ballistic and Space Targets. IEEE Transactions on Aerospace and Electronic Systems, 46(1): 96–119, 2010.
[3] X. Rong Li, Vesselin P. Jilkov. Survey of maneuvering target tracking: decision-based methods. Proceedings Volume 4728, Signal and Data Processing of Small Targets, 2002.
[4] Li, Xiao-Rong, Jilkov Vesselin P., A survey of maneuvering target tracking: approximation techniques for nonlinear filtering, Proceedings of the SPIE, Volume 5428, p. 537-550, 2004.
[5] X. Rong Li, V.P. Jilkov, Survey of maneuvering target tracking. Part V. Multiple-model methods, IEEE Transactions on Aerospace and Electronic Systems, 41(4): 1255-1321, 2005.

Reviewer 3 Report
The paper proposes a hybrid approach of UKF and Elman neural network to achieve adaptive tracking of maneuvering targets. Overall, the approach is interesting and sound. Yet my main concerns come from the evaluation results, which require some improvement in the design of simulation.
The formulation in Equation (1) indicates that the proposed approach is meant to address the kinematic model with dynamic model bias. However, it looks to me that this bias component did not exist in the evaluation setups in Section 4, which makes the system a linear one. Besides, I am not sure it is a fair comparison of results in Section 4 as the process noise covariance parameters used in Singer-UKF, IMM-KF, IM-ELM are not explained. Therefore, the results obtained from this are not convincing.
In line 212, it states that ENN is first trained using ground truth, then recursive learning and filtering starts following Equation (11). It looks like ground truth is mandatory to start the approach. I would like to ask how the approach deals with scenarios where obtaining ground truth is not possible, which is the case in most of real applications?
There are also a few minor issues in the paper.
Equation (12) implies that the state vector should be Xk|k = [x, vx, ax, y, vy, ay, z, vz, az]^T instead of the one in line 222.
The errors in Figure 5(b)(c)(d) are too hard to distinguish from one another. A clearer presentation is needed.
In Equation (9), predicted state X and covariance matrix P have a bar on top, which should be with a hat on top.
Lastly, the "experiments" are misleading as these are more of simulations.
While the topic and proposed approach are interesting, I would suggest a major revision to the manuscript, especially the evaluation results section.
Author Response
Response to Reviewer 3 Comments
Point 1: The formulation in Equation (1) indicates that the proposed approach is meant to address the kinematic model with dynamic model bias. However, it looks to me that this bias component did not exist in the evaluation setups in Section 4, which makes the system a linear one.
Response 1: In our experiments, the simulated target flight trajectories are generated based on the combination of five basic motion equations, such as line, parabola, serpentine and hyperbola and joint between them, at each trajectory segment or joint, where its acceleration is variable or with different configuration. An edit toolkit is used to generate any flights trajectories with maneuver. And in our experiments, for simplicity, the motion equation is modeled as the constant acceleration (CA) model, but the measurement function is a nonlinear equation, the whole tracking system is a nonlinear system also. According to this design, there is unknown and time-varying dynamic model bias between the trajectory and the CA equation, which is the maneuver item. Our algorithm can learn to track and filter more precision in this kind of problem.
To state more clearly, we added some explanation in the Line 274 to Line 279 as “In our experiments, the simulated target flight trajectories are based on the combination of five basic motion equations, such as line, parabola, serpentine and hyperbola and a joint between them, at each trajectory segment or joint, where its acceleration is variable or with different configuration. Ours trajectory edit toolkit can be used to generate any flights trajectories with maneuver. According to this design, there is unknown and time-varying dynamic model bias between the real trajectory and the CA equation, which is the maneuver item.”
Point 2: Besides, I am not sure it is a fair comparison of results in Section 4 as the process noise covariance parameters used in Singer-UKF, IMM-KF, IM-ELM are not explained. Therefore, the results obtained from this are not convincing.
Response 2: Sure, in order to indicate the compare is pair, the motion equations, measurement error, and initial parameters of process noise covariance Q, measurement noise covariance R, and state error covariance P are the same. And we select CA model for most methods in our paper.
To state more clearly, we added some explanation in the Line 286 to Line 287 as “In the comparison experiments, the motion equations and initial parameters of these covariance matrixes P, Q and R are set with the same parameters. And CA model is selected in these methods.”
Point 3: In line 212, it states that ENN is first trained using ground truth, then recursive learning and filtering starts following Equation (11). It looks like ground truth is mandatory to start the approach. I would like to ask how the approach deals with scenarios where obtaining ground truth is not possible, which is the case in most of real applications?
Response 3: In theory, if we can get the ground truth of target position and uncertainty of filter, the supervised training would be more perfect. However, when we track non-cooperative targets, it’s not practical. We have designed an approximate method, the feedback parameter and corrected state can be learnt online by a sliding-window scheme, where the tracking and learning is working simultaneously. And at last the Monte Carlo experiments show it works better and has good potential.
To state more clearly, we modified some explanation in the Line 228 to Line 235 as “We have designed an approximate method, the feedback parameter and corrected state can be learnt online by a sliding-window scheme. The training of the network is an online supervised training process and the training samples are collected by a sliding window with length N (for example, sample size N=20). In the initial phase, the ENN is trained by the ground truth, which is available approximately through generating samples like these …”
Point 4: Equation (12) implies that the state vector should be Xk|k = [x, vx, ax, y, vy, ay, z, vz, az]^T instead of the one in line 222.
Response 4: Yes, we have made a mistake. We have corrected state vector to Xk|k = [x, vx, ax, y, vy, ay, z, vz, az]^T.
Point 5: The errors in Figure 5(b)(c)(d) are too hard to distinguish from one another. A clearer presentation is needed.
Response 5: We have added more method’s comparison, and have plotted the results again to emphasis on the initial error distribution. Seen as in Figure 5.
Point 6: In Equation (9), predicted state X and covariance matrix P have a bar on top, which should be with a hat on top.
Response 6: Yes, there is a mistake. We have corrected it. Seen as in Line 198.
Point 7: Lastly, the "experiments" are misleading as these are more of simulations.
Response 7: All experiments are based on the statement of our method, and conclusions are based on the Monte Carlo experiments. All data and source code can be checked. Thank you.

Round 2
Reviewer 3 Report
My concerns in the last review have been addressed.